# A new pharmacodynamic approach to study antibiotic combinations against enterococci *in vivo*: Application to ampicillin plus ceftriaxone

**Ivone Jimenez-Toro**[1,2], **Carlos A. Rodriguez**[1,2]*, **Andres F. Zuluaga**[1,2], **Julian D. Otalvaro**[2], **Omar Vesga**[1,3]

**1** GRIPE, School of Medicine, University of Antioquia, Medellín, Colombia, **2** Integrated Laboratory of Specialized Medicine (LIME), School of Medicine, University of Antioquia, Medellín, Colombia, **3** Infectious Diseases Unit, Hospital Universitario San Vicente Fundación, Medellín, Colombia

* andres.rodriguez@udea.edu.co

**Data Availability Statement:** All relevant data are within the manuscript and its Supporting Information files.

## Abstract

The combination of ampicillin (AMP) and ceftriaxone (CRO) is considered synergistic against *Enterococcus faecalis* based on *in vitro* tests and the rabbit endocarditis model, however, *in vitro* assays are limited by the use of fixed antibiotic concentrations and the rabbit model by poor bacterial growth, high variability, and the use of point dose-effect estimations, that may lead to inaccurate assessment of antibiotic combinations and hinder optimal translation. Here, we tested AMP+CRO against two strains of *E. faecalis* and one of *E. faecium* in an optimized mouse thigh infection model that yields high bacterial growth and allows to define the complete dose-response relationship. By fitting Hill's sigmoid model and estimating the parameters maximal effect ($E_{max}$) and effective dose 50 ($ED_{50}$), the following interactions were defined: synergism ($E_{max}$ increase $\geq 2 \log_{10}$ CFU/g), antagonism ($E_{max}$ reduction $\geq 1 \log_{10}$ CFU/g) and potentiation ($ED_{50}$ reduction $\geq 50\%$ without changes in $E_{max}$). AMP monotherapy was effective against the three strains, yielding valid dose-response curves in terms of dose and the index $fT_{>MIC}$. CRO monotherapy showed no effect. The combination AMP+CRO against *E. faecalis* led to potentiation (59–81% $ED_{50}$ reduction) and not synergism (no changes in $E_{max}$). Against *E. faecium*, the combination was indifferent. The optimized mouse infection model allowed to obtain the complete dose-response curve of AMP+CRO and to define its interaction based on pharmacodynamic parameter changes. Integrating these results with the pharmacokinetics will allow to derive the PK/PD index bound to the activity of the combination, essential for proper translation to the clinic.

## Introduction

Enterococci are the third leading cause of hospital-associated infections [1]. They display intrinsic and acquired resistance to a wide variety of antibiotics in clinical use, including newer agents used to treat vancomycin resistant enterococcus (VRE) infections. In addition, compounds considered bactericidal against other Gram-positive cocci are usually

**Funding:** This project was funded by Minciencias (Colombian Ministry of Science and Technology), grant 111571149738 (AFZ, CAR) and 785 National PhD Scholarship 2017 (IJT), and by the University of Antioquia. www.minciencias.gov.co www.udea.edu.co The funders had no role in study design, data collection and analysis, decision to publish, or preparation of the manuscript.

**Competing interests:** I have read the journal's policy and the authors of this manuscript have the following competing interests: CAR has received honoraria for lectures on the therapeutic equivalence of generics and biosimilars from Allergan, Biosidus, Novartis and Pfizer, unrelated to this research project. AFZ has received honoraria for advisory boards and lectures on generics and biomilars therapeutic equivalence not related to the content of this paper from Allergan, Amgen, Janssen, Lilly, Merck, Novartis, Novo Nordisk, Pfizer, Roche and Sanofi. None of these companies or any other were involved in the design, execution, or publication of this study. IJT, JDO and OV have declared that no competing interests exist. This does not alter our adherence to PLOS ONE policies on sharing data and materials.

bacteriostatic against enterococci, posing a challenge for clinicians when faced with patients with severe infections [2–4] such as endocarditis and bacteremia, and often require combined therapies aiming for synergism [5–7]. One of these combinations is ampicillin (AMP) plus ceftriaxone (CRO).

The history of this dual β-lactam combination dates back to 1995 when Mainardi et al. reported an *in vitro* synergistic effect between amoxicillin (AMX) and cefotaxime (CTX) against several *Enterococcus faecalis* strains [8]. A few years later, Gavaldà et al. found *in vivo* synergism of AMP plus CRO against both high-level gentamicin resistant (HLGR) and aminoglycoside susceptible *E. faecalis* strains in the rabbit endocarditis model [9, 10]. Their findings were later confirmed in a clinical trial [11] and are the basis for the recommendation to use AMP+CRO for *E. faecalis* endocarditis, with the additional advantage of less nephrotoxicity compared to aminoglycoside-containing regimens [5].

Notwithstanding, the interaction of AMP and CRO is only understood partially, because most of the data have been derived from *in vitro* testing (checkerboard and time-kill curves) [8, 12] and animal models of endocarditis [9, 10, 13, 14] that have several limitations for PK/PD analysis and the optimal translation to humans: (i) deficient bacterial growth, (ii) low statistical power due to intrinsic high variation and, (iii) the lack of a complete dose-response curve due to the small number of doses tested [15, 16]. These limitations preclude the accurate estimation of pharmacodynamic parameters and the determination of the PK/PD index driving the efficacy of the combination, essential for proper extrapolation to the clinic [17, 18]. Additionally, the mortality rate of endocarditis has not changed during the last 30 years, suggesting that the there is room for improvement by optimizing drug combinations and dosing regimens based on the pharmacodynamics. [5, 6].

Our group developed an optimized neutropenic mouse thigh infection model of enterococci that yields bacterial growth in control animals of at least 2 $\log_{10}$ CFU/g in 24 h, and has allowed to estimate accurately the pharmacodynamic parameters of anti-enterococcal antibiotics in monotherapy fitting Hill's sigmoid model to the dose-response data [19]. Our aim here was to characterize the pharmacodynamics of the AMP+CRO combination in the optimized model and to analyze the interaction with a novel approach based on the parameters derived from Hill's equation. Partial results of this work were presented at ASM Microbe 2018 [20].

## Materials and methods

### Bacterial strains, antibiotics and susceptibility testing

For *in vitro* and *in vivo* studies, the strains *E. faecalis* ATCC 29212, *E. faecium* ATCC 19434 and *E. faecalis* ATCC 51299 (vancomycin-resistant, with VanB phenotype), were used and kept at −70˚C. *In vivo* experiments with *E. faecalis* ATCC 29212 were done with Ampicillin (Ampicilina, Genfar, Colombia) and Ceftriaxone (Rocephin, Roche, Switzerland). Due to previous reports of inequivalent ampicillin generics [21], the *in vivo* activity of the Genfar product (AMP innovator is not available in Colombia) was compared to innovator Ampicillin-Sulbactam (Unasyn, Pfizer, Switzerland) against β-lactamase negative enterococci and found to be equivalent in efficacy and potency. Due to later Genfar generic shortage, the experiments with *E. faecium* ATCC 19434 and *E. faecalis* ATCC 51299 were done with Ampicillin-Sulbactam. The minimal inhibitory concentrations of AMP and CRO were determined by broth microdilution in duplicate and repeated independently three times following CLSI methods [22].

### Time-kill curves (TKC)

Flasks containing Brain-Heart infusion (BHI) broth with AMP at 0.5, 1 and 2 times the MIC and/or CRO at a fixed concentration of 4 mg/L, were inoculated with a 6 $\log_{10}$ CFU/mL

bacterial suspension of *E. faecalis* ATCC 29212 and *E. faecium* ATCC 19434, prepared according to CLSI method [23]. Tubes containing only broth or bacteria without antibiotics were used as sterility and growth controls, respectively. After 0, 1, 2, 4, 8 and 24 h of incubation, aliquots of each culture (0.1 mL) were obtained. The samples were washed twice with sterile saline after centrifuging at 14,000 g for 10 minutes to prevent drug carryover. The final pellet was serially diluted and spread onto BHI agar plates and incubated for 24 h at 37˚C under aerobic atmosphere.

## *In vitro* drug interaction analysis

Individual time-kill curve data were analyzed by comparing the number of bacteria remaining after 24 hours of antibiotic exposure. By definition, synergism occurred when the combination killed 2 $log_{10}$ CFU/mL or more than the single most active drug. Antagonism when the combination killed at least 2 log CFU/mL less than the most active individual agent. In-between killing values indicated indifference [24].

## Inoculum preparation for *in vivo* experiments

The optimized mouse model requires that the inoculum is prepared under anaerobic conditions and with addition of porcine mucin. The protocol is described in detail in reference [19]. Briefly, the strains were recovered by two successive streaks on brain heart infusion (BHI) agar (Becton Dickinson, Heidelberg, Germany) with 5% sheep blood, followed by incubation for 24 h at 37˚C under anaerobic atmosphere (GazPak EZ; Becton Dickinson). For all strains, 3 colonies were suspended in 10 mL of thioglycolate USP broth (Oxoid, United Kingdom), serially diluted 4 times (1:10), and incubated overnight at 37˚C. The most diluted tube with complete turbidity was further diluted twice (1:10) in fresh broth, and incubated approximately for 3 hours, until it reached an $OD_{580nm}$ corresponding to ~8.0 $log_{10}$ CFU/mL. Finally, this tube was diluted 1:10 twice and mixed 50:50 with autoclaved 10% (wt/vol) porcine stomach mucin (Sigma-Aldrich, United States), yielding a 5.7 $log_{10}$ CFU/mL bacterial suspension with 5% mucin to inoculate the animals.

## Mice

Murine-pathogen free (MPF) Swiss albino mice of the strain Udea:ICR(CD-2), bred at the University of Antioquia MPF vivarium were used. They were fed and watered *ad libitum*, housed at a maximum density of 7 animals per box within a 693 $cm^2$ area in a One Cage System® (Lab Products, USA), and kept under controlled temperature (20˚C and 25˚C) and lightning conditions (12-hour day-night cycles). The personnel in charge of *in vivo* experiments was trained in animal care and the thigh infection model procedures (anesthesia, thigh inoculation, intraperitoneal and subcutaneous injections, euthanasia and tissue processing) by the senior researchers of the Infectious Diseases Problem Research Group (GRIPE, University of Antioquia). Animals were randomly picked and allocated to treatment or control groups (experimental units). The health condition of the mice was checked every day of the experiment and every 3 hours during the treatment phase (the last 24 hours). The following scale was used to classify the animals: 0: no signs of disease: active mouse, well groomed, alert, active; 1: mild signs of disease, as altered hair, slightly hunched posture with preserved mobility and response to stimuli; 2: moderate signs of disease, including squinted eyes, reduced mobility or reactivity, but able to reach water and food; 3: severe signs of disease, as great difficulty to reach water and food, dehydration (sunken eyes), reduced or no response to touch. If an animal reached phase 3 before the end of the experiment, it was sacrificed immediately by cervical dislocation under isoflurane anesthesia. No animal died before meeting the euthanasia criteria.

The study was reviewed and approved by the University of Antioquia Animal Experimentation Ethics Committee (July 9[th] 2015 session) and complied with the national guidelines for biomedical research (Resolution 008430 of 1993 by the Colombian Health Minister, articles 87 to 93) and the ARRIVE guidelines (S1 File).

### Optimized mouse infection model

We used our optimized murine thigh infection model for enterococci described in reference [19]. Six-week-old female mice weighing 23 to 27 g were rendered neutropenic by two intraperitoneal injections of cyclophosphamide (Endoxan; Baxter, Germany), given four days (150 mg/kg of body weight) and one day (100 mg/kg) before infection [25]. The mice were inoculated in each thigh with 100 μL of the bacterial suspension described above under isoflurane anesthesia. Treatment started 2 h post-infection and lasted 24 h (the total duration of the experiment from the first cyclophosphamide injection to the end of antibiotic treatment was 6 days). Mice were allocated in groups of two to receive monotherapy with AMP at doses from 9.4 to 2400 mg/kg/day and CRO from 3.125 to 200 mg/kg/day (5 to 6 doses per experiment). For AMP+CRO combined therapy, the complete range of AMP doses were administered in combination with one fixed dose of CRO: 3.125, 12.5, 25, 50, 100 or 200 mg/kg/day (2 mice per dose, 5 to 8 doses per experiment). The antibiotics were injected subcutaneously every 3 h (200 μL per injection). At the end of treatment, the animals were euthanized by cervical dislocation under isoflurane anesthesia, the thighs were dissected aseptically, homogenized, serially diluted, plated, and incubated overnight for bacterial quantification (limit of detection: 2 $\log_{10}$ CFU/g). Groups of two untreated but infected control mice were euthanized at the initiation (0 hour) and at the end of the treatment (24 hours) to assess bacterial growth. Six independent experiments were done with *E. faecalis* ATCC 29212 (138 mice), one with *E. faecium* ATCC 19434 (32 mice) and one with *E. faecalis* ATCC 51299 (40 mice).

### *In vivo* data analysis

The net antibacterial effect of each antibiotic dose was calculated by subtracting the number of CFU/g in untreated controls at 24 h from the number of CFU/g remaining in treated mice. Least-squares nonlinear regression was used to fit Hill's Equation to the dose-effect data and estimate the primary parameters maximum effect ($E_{max}$, to quantify efficacy), 50% effective dose ($ED_{50}$, to quantify potency), and Hill's slope (N). Additionally, the secondary parameters bacteriostatic dose (BD), 1-$\log_{10}$ kill dose (1LKD) and 2-$\log_{10}$ kill dose (2LKD) were calculated. The differences between the complete dose-response curves or the individual parameters were tested by curve-fitting analysis (CFA). Regressions was assessed by the adjusted coefficient of determination (Adj.$R^2$) and the standard error of estimate ($S_{y.x}$) and were considered valid only if they passed the normality of residuals and homoscedasticity tests. The experiments could be analyzed separately or jointly if a single regression curve described the datasets better than individual ones, as indicated by the overall test for coincidence (extra sum-of-squares F test). All analyses were run in Prism 7.0 (GraphPad, San Diego, United States) [26, 27].

### PK/PD analysis

We used a 2-compartment pharmacokinetic model of AMP in infected ICR mice estimated by nonparametric techniques in a previous study [19]. The median of the parameters were: $K_{el}$ (elimination rate constant) 20.4 h[-1], $V_c$ (volume of the central compartment) 0.007 L, $K_{cp}$ (transfer rate constant from the central to the peripheral compartment) 78.2 h[-1], $K_{pc}$ (transfer rate constant from the peripheral to the central compartment) 15.0 h[-1], and $K_a$ (absorption rate constant) 6.45 h[-1] [19, 21]. AMP exposure in the monotherapy studies was calculated in

terms of the time that the free drug concentration exceeded the MIC ($f\text{T}_{>\text{MIC}}$) for each of the doses and strains using Monte Carlo simulation in the Pmetrics package for R, including the between-subject variability of the parameters (S1 Table) [28]. The PK/PD index was plotted against the antibacterial effect to estimate the magnitudes required for stasis (BD), 1-$\log_{10}$ kill (1LKD) and 2-$\log_{10}$ kill (2LKD). AMP protein binding was not considered because it is only 3% in mice [29].

### *In vivo* interaction analysis

Drug interaction was analyzed taking into account the complete dose-response curve derived by a valid nonlinear regression fitting Hill's equation. Based on the changes of the PD parameter values with the combination, four terms of interaction were used: synergism, potentiation, antagonism and indifference. Synergism was defined as an $E_{max}$ increase of at least 2 $\log_{10}$ CFU/g, potentiation as an $ED_{50}$ reduction $\geq$50% without significant changes in $E_{max}$, and antagonism as an $E_{max}$ reduction of 1 $\log_{10}$ CFU/g or more in the combination compared to the single most effective drug. Changes below these thresholds, even if statistically significant, were considered indifferent. Parameter differences were assessed by Curve Fitting Analysis (CFA) (Prism 7.0).

### Human AMP pharmacokinetics/pharmacodynamics simulation

To put the animal data in clinical context we simulated the human pharmacokinetics of two AMP doses with ADAPT 5 [30]: 500 mg every 6 hours (for soft tissue infections) and 2000 mg every 4 hours (for enterococcal endocarditis), both administered in 15-minute intravenous bolus, based on the population 2-compartment model published by Soto et al. [31] with the following parameters for a typical patient weighing 70 kg and with a creatinine clearance of 71 mL/min: total clearance ($CL_t$) 10.7 L/h, volume of the central compartment ($V_c$) 9.97 L, distributional clearance ($CL_d$) 4.48 L/h and volume of the peripheral compartment ($V_p$) 6.14 L, and estimated the percentage free time above concentrations ranging from 0.5 to 256 mg/L, with 20% protein binding.

## Results

### Susceptibility testing and time-kill curves

The three strains were AMP-susceptible and CRO-resistant (CRO is inactive in monotherapy against enterococci). The modal MIC values are shown in Table 1. In the time kill curves, growth at 24 h in the CRO group was almost the same as the controls as expected. The combination of AMP at concentrations of 1 mg/L and higher with CRO fixed at 4 mg/L was synergistic against *E. faecalis* ATCC 29212, increasing the bacterial killing 4.94 $\log_{10}$ CFU/mL at AMP 1 mg + CRO, and 2.62 $\log_{10}$ CFU/mL at AMP 2 mg/L + CRO compared with AMP alone (Fig 1 panel a). Against *E. faecium* ATCC 19434 the combination was indifferent: at AMP 0.5 mg/L + CRO, bacterial killing increased only 0.57 $\log_{10}$ CFU/mL, and at AMP 1 mg/L + CRO, only 0.22 $\log_{10}$ CFU/mL (Fig 1 panel b).

### *In vivo* pharmacodynamics

The bacterial burden in the thighs of untreated mice increased in average 2.02, 2.05, and 3.42 $\log_{10}$ CFU/g after 24 h for *E. faecalis* 29212, *E. faecalis* 51299 and *E. faecium* 19434, respectively, the expected growth of enterococci in the optimized thigh model. Hill's equation fitted well to the data, yielding valid regressions and statistically significant parameters for AMP monotherapy and all the AMP+CRO combinations (Table 2). CRO in monotherapy was tested

**Table 1. Minimal inhibitory concentrations of three strains of enterococci.**

| Antibiotic | CLSI Reference.[b] | MIC (modal value in mg/L) [a] | | |
|---|---|---|---|---|
| | | *E. faecalis* ATCC 29212 | *E. faecalis* ATCC 51299 (VanB) | *E. faecium* ATCC 19434 |
| Ampicillin | 0.5–2 | 1 | 0.5 | 0.5 |
| Ceftriaxone | NA | 128* | 256* | 128* |
| Gentamicin | 4–16 | 8 | >256 | 8 |
| Linezolid | 1–4 | 2 | 2 | 8 |
| Moxifloxacin | 0.06–0.5 | 0.25 | 0.25 | 2 |
| Tigecycline | 0.03–0.12 | 0.5 | 0.5 | 0.5 |
| Vancomycin | 1–4 | 2 | 64 | 4 |
| Daptomycin | 1–4 | 2 | 2 | 4 |

a. Obtained from three independent duplicate assays.

b. For *E. faecalis* ATCC 29212 (control strain) by standard broth microdilution.

NA, not applicable

* There is no CLSI breakpoint for ceftriaxone against enterococci.

only against *E. faecalis* ATCC 29212 and was completely ineffective (bacterial burden without difference to control mice, S1 Fig).

Regarding the PK/PD analysis of AMP monotherapy, the $f$T$_{>MIC}$ necessary for bacteriostasis and killing varied according to the strain. In the case of *E. faecalis* 29212, the magnitude required for stasis was 28.7%, 33.3% for 1-log$_{10}$ kill and 39.9% for 2-log$_{10}$ kill (Fig 2 and Table 3). Against *E. faecalis* 51299, the BD was 35.8% and the 2LKD 67.2% (Table 3, S1 Fig). With *E. faecium* 19434, its higher growth required longer exposures for stasis and killing but with minimal difference between BD and 2LKD (51.1% and 52.8%, respectively) due to the steep slope of the dose-response curve (Table 3, S2 Fig). The PK/PD analysis of AMP+CRO was not done because the index linked to the efficacy of this combination has not been defined.

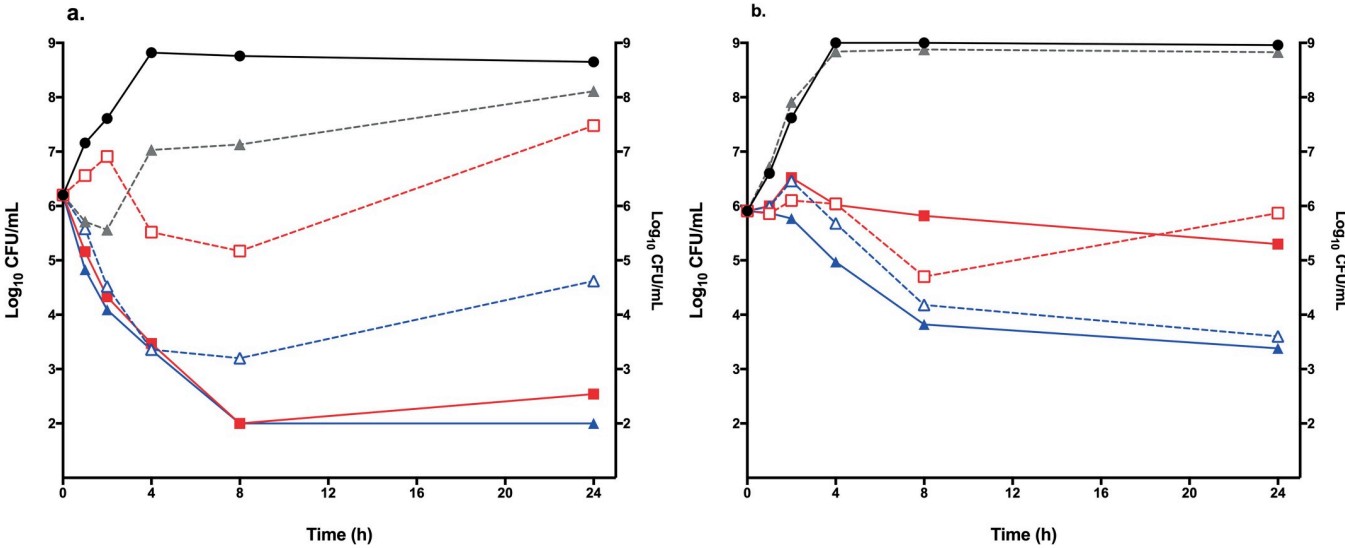

**Fig 1. *In vitro* time-kill curves of AMP+CRO. (a)** *E. faecalis* ATCC 29212, **(b)** *E. faecium* ATCC 19434. Growth control (solid black line and circles), CRO 4 mg/L (gray dashed line and triangles), AMP 1xMIC (red dashed line and squares), AMP 1xMIC plus CRO 4 mg/L (solid red line and squares), AMP 2xMIC (blue dashed line and triangles) and AMP 2xMIC plus CRO 4 mg/L (solid blue line and triangles).

**Table 2.** *In vivo* pharmacodynamics of AMP and AMP plus CRO against enterococci.

| Strain | Treatment (CRO mg/kg/day) | Parameter magnitude ± SE | | | Adjusted $R^2$ | $S_{y.x}$ ($log_{10}$ CFU/g) | P value by CFA ($ED_{50\_AMP}$ vs. $ED_{50\_AMP+CRO}$) [a] |
|---|---|---|---|---|---|---|---|
| | | $E_{max}$ ($log_{10}$ CFU/g) | Hill's slope (N) | $ED_{50}$ (mg/kg/day) | | | |
| *E. faecalis* ATCC 29212 | AMP | 4.96 ± 0.17 | 2.35 ± 0.41 | 126.1 ± 11.1 | 0.93 | 0.53 | NA |
| | AMP + CRO (200) | 5.55 ± 0.2 | 1.33 ± 0.2 | 46.7 ± 6.17 | 0.97 | 0.33 | 0.0004 |
| | AMP + CRO (50) | 5.27 ± 0.19 | 1.59 ± 0.27 | 52.7 ± 5.98 | 0.93 | 0.43 | <0.0001 |
| | AMP + CRO (37.5) | 4.86 ± 0.15 | 2.18 ± 0.44 | 51.6 ± 5.28 | 0.98 | 0.29 | 0.0002 |
| | AMP + CRO (25) | 4.78 ± 0.22 | 2.1± 0.60 | 51.9 ± 8 | 0.95 | 0.42 | 0.0004 |
| | AMP + CRO (12) | 4.34 ± 0.20 | 3.96 ± 4.6 | 117.8 ± 33.9 | 0.97 | 0.41 | 0.9105 |
| | AMP + CRO (3.12) | 4.34 ± 0.14 | 2.75 ± 0.66 | 109 ± 11 | 0.96 | 0.39 | 0.4747 |
| *E. faecium* ATCC 19434 | AMP | 7.58 ± 0.29 | 5.82 ± 1.14 | 239.4 ± 8.26 | 0.97 | 0.56 | NA |
| | AMP + CRO (200) | 7.65 ± 0.31 | 3.81 ± 0.68 | 197.6 ± 10.1 | 0.97 | 0.58 | 0.0067 |
| *E. faecalis* ATCC 51299 | AMP | 4.77 ± 0.30 | 1.1 ± 0.17 | 128.3 ± 25.4 | 0.96 | 0.3 | NA |
| | AMP + CRO (100) | 4.88 ± 0.23 | 0.88 ± 0.18 | 25.7 ± 4.16 | 0.92 | 0.29 | 0.0019 |
| | AMP + CRO (25) | 4.50 ± 0.07 | 1.42 ± 0.12 | 24.5 ± 1.26 | 0.98 | 0.14 | <0.0001 |

Abbreviations: AMP, ampicillin; CRO, ceftriaxone; $E_{max}$, maximum effect; $ED_{50}$, 50% effective dose; N, Hill's slope; $S_{y.x}$: standard error of estimate, NA, not applicable.

[a] $ED_{50}$ magnitudes were compared by curve fitting analysis (CFA).

All values are presented as means and standard errors.

### *In vivo* interaction analysis

The dose-response curves of AMP monotherapy and all the AMP+CRO combinations against *E. faecalis* 29212 are shown in Fig 3. The addition of CRO at doses of 3.125 and 12.5 mg/kg/day produced no significant effect and yielded the same curve of AMP monotherapy (P = 0.1427). However, starting at 25 mg/kg/day of CRO and up to 200 mg/kg/day, AMP $ED_{50}$ was reduced >50% without significant changes in $E_{max}$ (Table 2). Notably, these treatment groups (AMP+CRO 25, 37.5, 50 and 200 mg/kg/day) were better described by a single regression (P = 0.1029), yielding an $E_{max}$ of 5.15 ± 0.10 $log_{10}$ CFU/g (P = 0.3164 vs. AMP monotherapy) and $ED_{50}$ of 51.1 ± 3.42 mg/kg/day (P<0.0001 vs. AMP monotherapy, a 59% reduction), a case of potentiation (Fig 4).

Against *E. faecalis* 51299, AMP potency was also increased when combined with CRO at doses of 25 mg/kg/day and higher without changing efficacy ($ED_{50\_AMP}$ 128.3 ± 25.4 vs. $ED_{50\_AMP+CRO}$ 24.5 ± 1.26 mg/kg/day; a reduction of 81%, P = <0.0001), indicating also potentiation (Fig 5 panel a). Finally, AMP in monotherapy was highly bactericidal against *E. faecium*, reaching a ~5 $log_{10}$ CFU/g bacterial reduction after 24 hours of treatment (Fig 5 panel b). In combination, the $ED_{50}$ was significantly reduced ($ED_{50\_AMP}$ 239.4 ± 8.24 vs. $ED_{50\_AMP+CRO}$ 197.7 ± 10.05 mg/kg/day, P = 0.0067), but only 17%, thus it did not reach the threshold for potentiation and the interaction was indifferent.

### Human AMP pharmacokinetics/pharmacodynamics simulation

The simulated PK profile of AMP is shown in S4 Fig (500 mg q6h) and S5 Fig (2000 mg q4h). With the lower dose, the total $C_{max}$ was 42 mg/L (34 mg/L free) and the $C_{min}$ 0.9 mg/L (0.7 mg/L free). With the higher dose the total $C_{max}$ and $C_{min}$ were 176 and 10 mg/L, respectively (corresponding to free concentrations of 141 and 8 mg/L). The time that the free concentration of AMP was above the MICs is displayed in S2 Table. With 500 mg q6h dose the $fT_{>MIC}$ was at least 39% with MICs up to 4 mg/L, with the 2000 mg q4h dose the $fT_{>MIC}$ was at least 37% with MICs up to 32 mg/L.

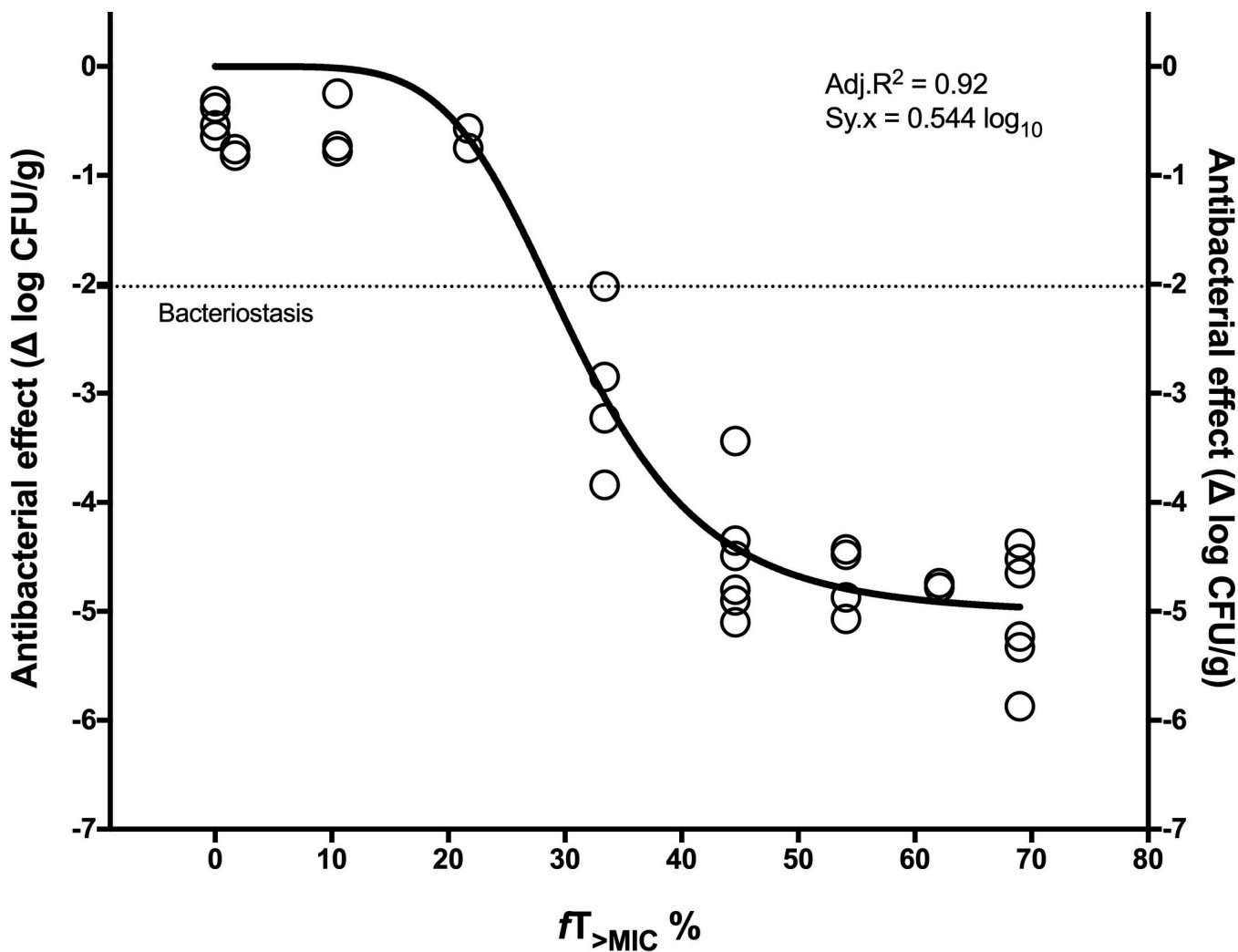

**Fig 2. Pharmacodynamics of AMP monotherapy against *E. faecalis* ATCC 29212 in terms of $f$T$_{>MIC}$.** The data are derived from three independent experiments (N = 46 mice) that were undistinguishable by curve-fitting analysis (i.e. a single curve described better the three data sets than independent ones). BD was 28.7%, 1LKD 33.3% and 2LKD 39.9% (Table 3).

## Discussion

We found that the combination of AMP and CRO is synergistic *in vitro* (determined by time-kill curves) against *E. faecalis*, as previously reported and explained by the saturation of multiple PBPs: PBPs 4 and 5 by AMP and PBPs 2 and 3 by CRO, leading to increased kill [8, 9]. In

**Table 3. BD, 1LKD and 2LKD of AMP vs. enterococci in terms of $f$T$_{>MIC}$.**

| | $f$T$_{>MIC}$ (%) * | | |
|---|---|---|---|
| **Parameter** | ***E. faecalis* ATCC 29212** | ***E. faecalis* ATCC 51299 (VanB)** | ***E. faecium* ATCC 19434** |
| BD | 28.7 ± 1.46 | 35.8 ± 2.12 | 51.1 ± 0.86 |
| 1LKD | 33.3 ± 1.25 | 42.9 ± 2.75 | 51.9 ± 0.1 |
| 2LKD | 39.9 ± 1.87 | 67.2 ± 6.73 | 52.8 ± 0.13 |

Abbreviations: BD, bacteriostatic dose; 1LKD, dose required to kill 1-log$_{10}$ CFU/g; 2LKD, dose required to kill 2-log$_{10}$ CFU/g. All values are presented as means standard errors.

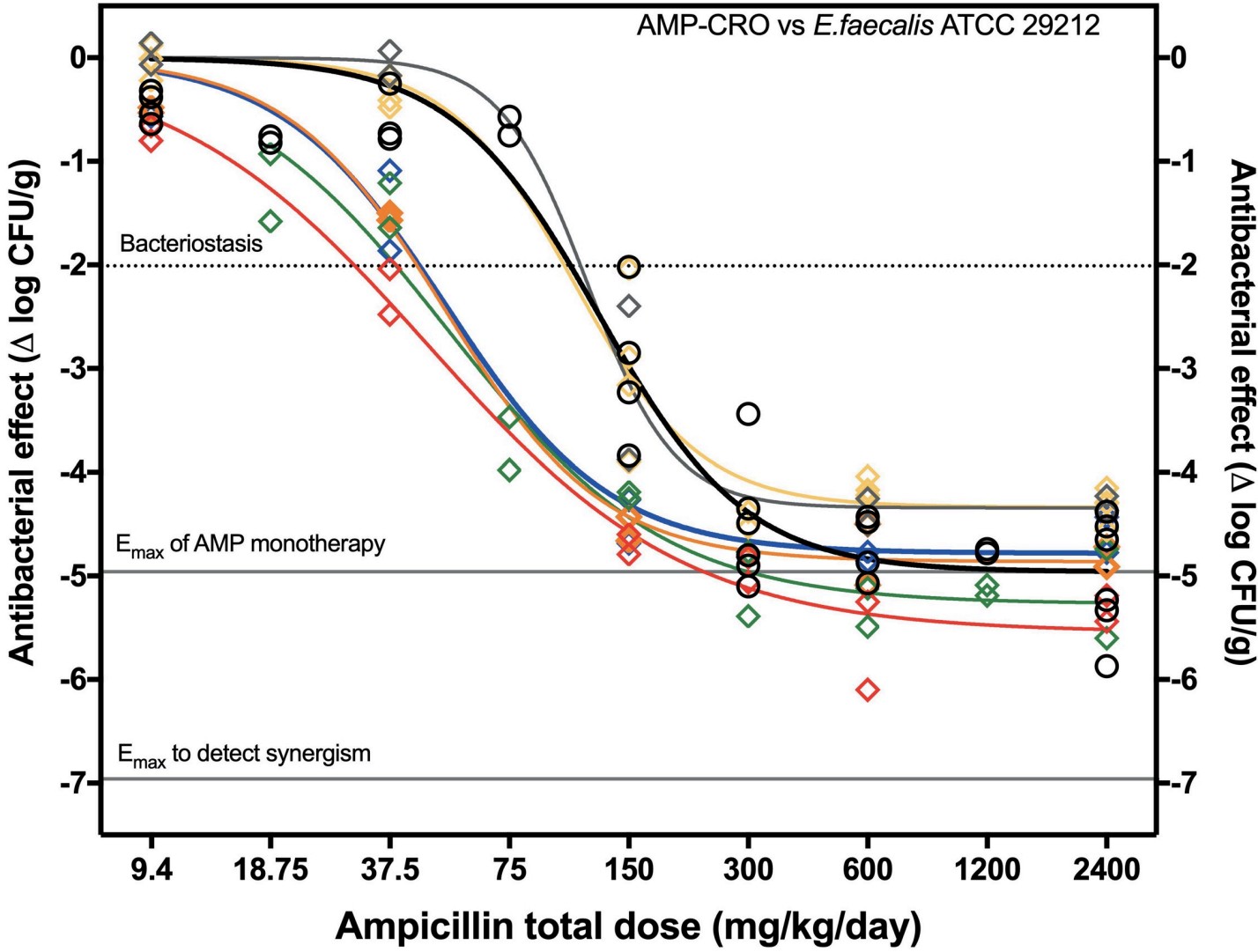

**Fig 3. Dose-response curves of AMP monotherapy and AMP+CRO combinations vs. *E. faecalis* ATCC 29212.** AMP Monotherapy (black), AMP+CRO 200 (red), AMP+CRO 50 (green), AMP+CRO 37.5 (orange), AMP+CRO 25 (blue), AMP+CRO 12.5 (gray) and AMP+CRO 3.125 (yellow). The PD parameters are presented in Table 2.

contrast, the combination was indifferent against *E. faecium* ATCC 19434, consistent with published data of non-uniform synergism (isolate-dependent) against this enterococcal species [32].

*In vivo*, using the optimized mouse thigh infection model, we were able to obtain a complete dose-response curve of AMP monotherapy, with valid parameters derived from Hill's sigmoid model to quantify its efficacy and potency, and the magnitude of the PK/PD index that drives its activity ($fT_{>MIC}$) for stasis, 1 $\log_{10}$ kill and 2 $\log_{10}$ kill (something not possible with the previous models characterized by poor growth and high variability). We also obtained valid response-curves of AMP combined with several doses of CRO, and by assessing the changes in the PD parameters it was determined that *in vivo*, the interaction against *E. faecalis* was potentiation (>50% reduction of $ED_{50}$ without changes in $E_{max}$) and not synergism (increased maximal effect), in apparent discordance with the results of the rabbit endocarditis models that showed increased killing (i.e. synergism), and the clinical trials.

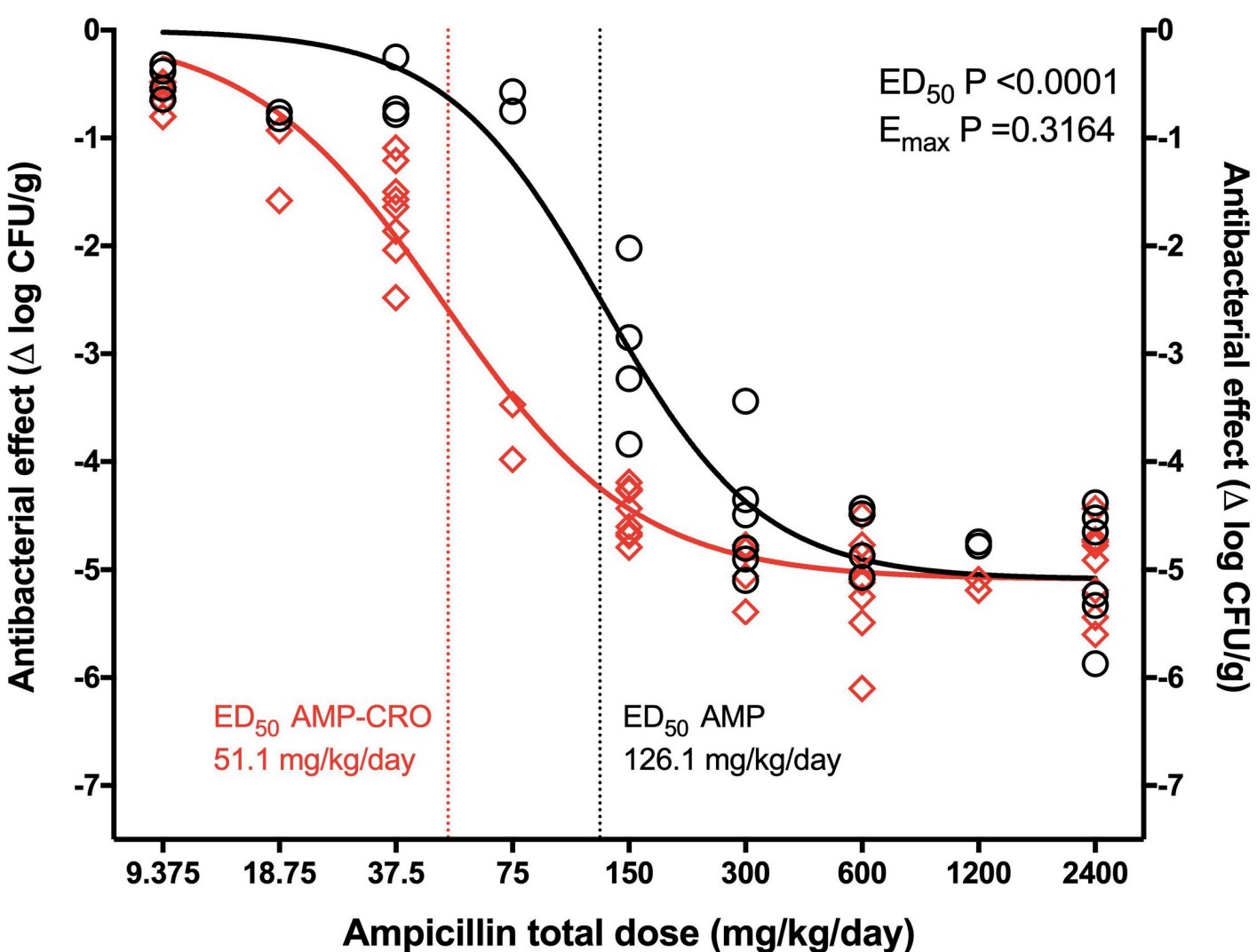

**Fig 4. Dose-response curves of AMP monotherapy and combined AMP+CRO vs. *E. faecalis* ATCC 29212.** AMP monotherapy (black), AMP+CRO 25, 37.5, 50 and 200 mg/kg/day combined in a single regression (red). There was no difference in $E_{max}$ but a significant 59% reduction in the $ED_{50}$ with the combination, indicating potentiation.

The key to elucidate this apparent discordance may be in the shape of the dose-response curve and the position of a specific dose in it. When there is potentiation, the monotherapy and combination curves converge at the extremes (no efficacy and maximal efficacy regions), but they separate in the middle (the combination curve moves to the left of the monotherapy one, as seen in Fig 4), then, if the ampicillin dose used is located beyond the point of maximal efficacy (plateau), the addition of ceftriaxone will sum nothing to bacterial killing and the interaction will be interpreted as indifferent. However, if the dose is in the middle to left region of the curve, ceftriaxone will increase bacterial killing by several $\log_{10}$ CFU/g, turning a bacteriostatic dose into a bactericidal one, and the interaction will be interpreted as synergism. Then, in studies that use a single point of the dose-response curve, the interaction is defined according to the specific effect at that point, overlooking the complete curve and potentially leading to discordant interpretations ("missing the forest for the trees") [33].

Regarding the comparison of murine and human exposures in terms of free time above MIC, the simulation with the ampicillin dose used for human soft tissue infections (500 mg

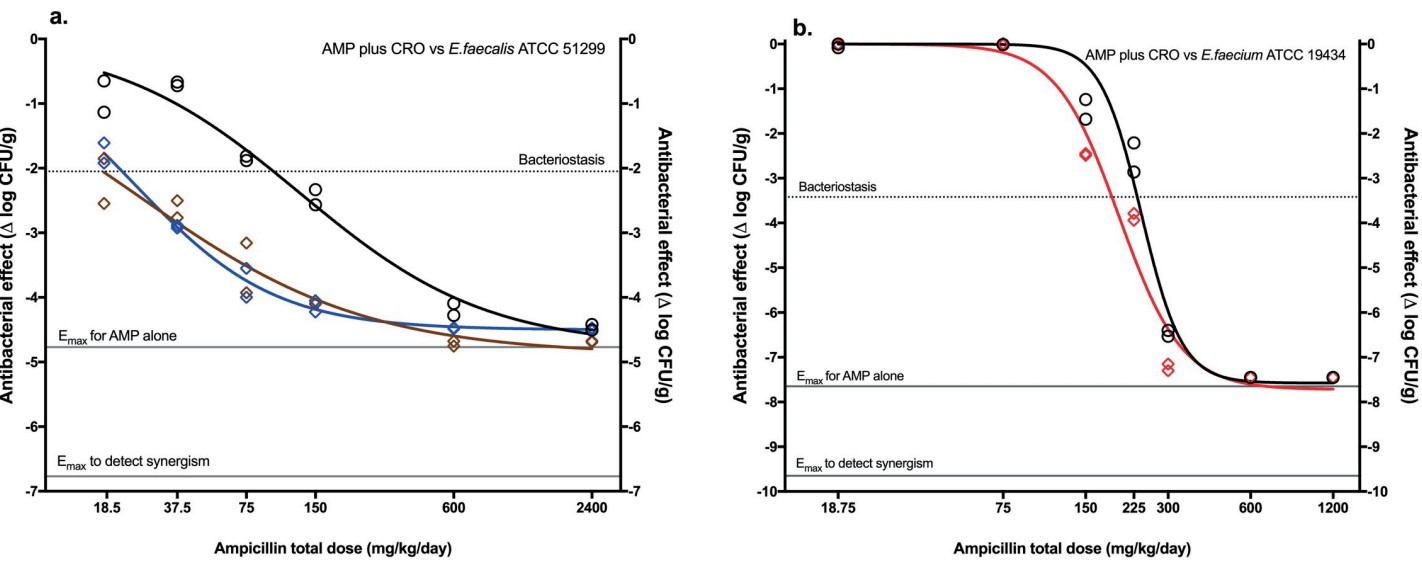

**Fig 5.** *In vivo* **dose-response curves of AMP and AMP+CRO against other strains of enterococci. (a)** *E. faecalis* ATCC 51299. **(b)** *E. faecium* ATCC 19434. AMP monotherapy (black), AMP+CRO 25 (blue) AMP+CRO 100 (brown) and AMP+CRO 200 (red). Parameter values are in Table 2.

q6h) showed that for strains with MICs up to 2 mg/L, the $f\text{T}_{>\text{MIC}}$ is at least 61.8%, an exposure leading to maximal efficacy and without benefit from CRO addition. However, in infections with strains exhibiting higher MICs (8 mg/L), this dose is located in the left region of the curve ($f\text{T}_{>\text{MIC}}$ 23.4%), below the required exposure for bacteriostasis, but sensitive to CRO potentiation. Thus, the addition of ceftriaxone would turn an ineffective AMP dose into a bactericidal one and the combination may allow to treat more resistant Enterococci without increasing the penicillin dose.

On the other hand, the high dose used for the treatment of endocarditis yields $f\text{T}_{>\text{MIC}}$ of at least 63% for strains with MICs up to 16 mg/L (S2 Table), suggesting that for the majority of Enterococcal isolates, this dose would be located in the right region of the curve, where no potentiation occurs. However, endocarditis is a difficult-to-cure infection due to (i) poor penetration of some antibiotics into the infected vegetations; (ii) altered metabolic state of bacteria within the lesion; and (iii) absence of adequate host-defense cellular response; also, endocarditis has been shown to require a higher antibiotic exposure for cure than non-endocardial infections. In the case of β-lactams, maintaining the concentration just above the MIC is not sufficient to ensure sterilization of vegetations [34], and data from the rabbit endocarditis model indicate that successful treatment requires through levels at least ten times the minimal bactericidal concentration (MBC). Considering that the ampicillin $\text{MBC}_{90}$ of *E. faecalis* has been reported to be >128 mg/L [35] and the $f\text{C}_{\text{max}}$ with 2000 mg q4h is around 140 mg/L, this dose would be actually located in the left (ineffective) region of a hypothetical endocarditis exposure-response curve (yet to be experimentally determined), and this would explain the increased killing effect observed with the CRO combination in the rabbit model and the results of the clinical trials.

These hypotheses deserve further study, but a PK/PD index linked specifically to the effect of the combination, not only the monotherapy, is required in order to better extrapolate to other animals and humans and to design optimal treatment regimens [36]. To date that index has not been identified, but there are some insights from this and previous work: here we found that the potentiation ($\text{ED}_{50}$ reduction) of AMP vs. *E. faecalis* ATCC 29212 began at a CRO dose of 25 mg/kg/day and did not change with increasing doses up to 200 mg/kg/day. At

the 12.5 mg/kg/day it was lost and the $ED_{50}$ was the same of AMP monotherapy. This quantal dose-effect relationship (all or none) suggests that there is a CRO exposure threshold for potentiation (in this case a dose between 12.5 and 25 mg/kg/day), and once it is reached, the $ED_{50}$ does not change despite dose increments. As CRO at a concentration of 4 mg/L reduces the MIC of AMP two to four-fold [9], this combination may be analogous to the penicillin-β-lactamase inhibitor combinations, where the inhibitor reduces the MIC of the partner β-lactam (i.e. a potentiation effect), and the PK/PD index driving the efficacy is the free time above a threshold ($fT_{>threshold}$), as described earlier for piperacillin-tazobactam [37, 38]. Also, considering that the ampicillin MIC varies continuously depending on the changing ceftriaxone concentration *in vivo*, a more accurate PK/PD index could be the time above the instantaneous MIC, as described by Bhagunde et al. for β-lactam-β-lactamase inhibitor combinations [39]. These ideas merit further development and will be elaborated in a separate paper.

In conclusion, this study presents the first *in vivo* pharmacodynamic evaluation of an anti-enterococcal drug combination with the optimized mouse thigh infection model and proposes a new method to study *in vivo* antibiotic interactions that relies on the estimation of the complete dose-response relationship (in contrast to other methods that use point dose-effect or concentration-effect measurements) and defines the type of interaction according to the changes in the PD parameters $E_{max}$ (synergism or antagonism) or $ED_{50}$ (potentiation). With the new method we showed that the AMP+CRO combination, considered synergistic against *E. faecalis* in *in vitro* studies and the rabbit endocarditis model, is really a case of potentiation in the murine thigh infection model, and the actual antibacterial effect will depend on the dose location in the exposure-response curve. This is also the first step for determining the PK/PD index linked to the combination effect, an essential requirement to translate the results to humans and to allow the optimization of dosing regimens. The method will be now tested with other old and new anti-enterococcal combinations.

## Supporting information

**S1 Fig. *In vivo* pharmacodynamics of CRO monotherapy vs. *E. faecalis* ATCC 29212.** CRO monotherapy was completely ineffective against *E. faecalis* in doses up to 200 mg/kg/day. Regression line is shown dotted because Hill's sigmoid model did not fit to the data.
(DOCX)

**S2 Fig. *In vivo* pharmacodynamics of AMP monotherapy vs. *E. faecalis* ATCC 51299.** The parameters BD, 1LKD and 2LKD are shown in Table 3 of the paper.
(DOCX)

**S3 Fig. *In vivo* pharmacodynamics of AMP monotherapy vs. *E. faecium* ATCC 19434.** The parameters BD, 1LKD and 2LKD are shown in Table 3 of the paper.
(DOCX)

**S4 Fig. Simulated PK profile of intravenous AMP 500 mg every 6 hours.** PK profile in a typical patient weighing 70 kg and with a creatinine clearance of 71 mL/min. The total serum concentration of AMP along 24 hours is displayed (protein binding is 20%).
(DOCX)

**S5 Fig. Simulated PK profile of intravenous AMP 2000 mg every 4 hours.** PK profile in a typical patient weighing 70 kg and with a creatinine clearance of 71 mL/min. The total serum concentration of AMP along 24 hours is displayed (protein binding is 20%).
(DOCX)

**S1 Table.** $f$T$_{>\mathbf{MIC}}$ **of AMP doses used against enterococci in mice.**
(DOCX)

**S2 Table. %** $f$T$_{>\mathbf{MIC}}$ **of AMP doses used against enterococci in humans.**
(DOCX)

**S1 File. ARRIVE guidelines checklist.**
(PDF)

**S2 File.** *In vivo* **raw data.**
(XLSX)

## Acknowledgments

We want to thank the Infectious Diseases Problems Research Group (GRIPE) and the Pharmacology and Toxicology Laboratory for their support during the experimental procedures, and Corporación Ciencias Básicas Biomédicas of the University of Antioquia for its academic support.

## Author Contributions

**Conceptualization:** Carlos A. Rodriguez, Andres F. Zuluaga, Omar Vesga.

**Data curation:** Ivone Jimenez-Toro, Carlos A. Rodriguez.

**Formal analysis:** Ivone Jimenez-Toro, Carlos A. Rodriguez, Julian D. Otalvaro, Omar Vesga.

**Funding acquisition:** Andres F. Zuluaga.

**Investigation:** Ivone Jimenez-Toro, Carlos A. Rodriguez.

**Methodology:** Ivone Jimenez-Toro, Carlos A. Rodriguez, Omar Vesga.

**Project administration:** Andres F. Zuluaga.

**Resources:** Andres F. Zuluaga.

**Software:** Julian D. Otalvaro.

**Supervision:** Carlos A. Rodriguez.

**Validation:** Carlos A. Rodriguez.

**Visualization:** Ivone Jimenez-Toro.

**Writing – original draft:** Ivone Jimenez-Toro.

**Writing – review & editing:** Ivone Jimenez-Toro, Carlos A. Rodriguez, Andres F. Zuluaga, Julian D. Otalvaro, Omar Vesga.

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
