## [Decision Letter · Decision Letter 0]

21 Sep 2020

PONE-D-20-19379

A new pharmacodynamic approach to study antibiotic combinations against Enterococci in vivo: application to Ampicillin plus Ceftriaxone

PLOS ONE

Dear Dr. Rodriguez,

Thank you for submitting your manuscript to PLOS ONE. After careful consideration, we feel that it has merit but does not fully meet PLOS ONE’s publication criteria as it currently stands. Therefore, we invite you to submit a revised version of the manuscript that addresses the points raised during the review process.

Your manuscript has been reviewed by two experts in your field.  A minor revision is suggested before a decision can be made.

We look forward to receiving your revised manuscript.

Kind regards,

Yung-Fu Chang

Academic Editor

PLOS ONE

Journal Requirements:

"I have read the journal's policy and the authors of this manuscript have the following

competing interests:CAR has received honoraria for lectures on the therapeutic

equivalence of generics and biosimilars from Allergan, Biosidus, Novartis and Pfizer,

unrelated to this research project. AFZ has received honoraria for advisory boards and

lectures on generics and biomilars therapeutic equivalence not related to the content of

this paper from Allergan, Amgen, Janssen, Lilly, Merck, Novartis, Novo Nordisk, Pfizer,

Roche and Sanofi. None of these companies or any other were involved in the design,

execution, or publication of this study. IJT, JDO and OV have declared that no

competing interests exist."

Reviewers' comments:

Reviewer's Responses to Questions

**Comments to the Author**

1. Is the manuscript technically sound, and do the data support the conclusions?

Reviewer #1: Yes

Reviewer #2: Yes

2. Has the statistical analysis been performed appropriately and rigorously? 

Reviewer #1: Yes

Reviewer #2: Yes

3. Have the authors made all data underlying the findings in their manuscript fully available?

Reviewer #1: Yes

Reviewer #2: Yes

4. Is the manuscript presented in an intelligible fashion and written in standard English?

Reviewer #1: Yes

Reviewer #2: Yes

5. Review Comments to the Author

Reviewer #1: interesting pkpd study of combination therapy for Enterococcus. The authors applied proven pkpd concepts and the study is well done. My only comment to the authors is that I agree that it is clear that combination effect is most pronounced in the middle of the exposure curve, and perhaps not surprisingly as they note that at the extremes of exposures, the addition of a second agent is unlikely a priori to make too much of a difference. so with that in mind, it begs a clear question here of where on the ampicillin exposure curve are the mice relative to human exposures, so that one can put this all in clinical context. For example, are the ampicillin exposures, given it is given IV at high doses and frequent administration, already maxing out at T>MIC >50%, and if so then this model does not explain why ceftriaxone may be helpful as when ampicillin exposures are >50% you max out the effect.

The only other comment I have is that another factor that is not taken into account in these studies is time. These are 24 h studies, you can only have so much antibacterial effect in 24h of time, and thus you might see potentiating or synergistic activities outside this 24h window. Many studies have shown that when you compare short and longer duration studies, you have a very similar ED50 but significantly change the Emax and slope as you get much deeper kill with longer duration. This is a good example that it does significantly depend where on the exposure curve you are located when translating over the animal model hill curves to clinical medicine (and again why elucidating where on the curve the human exposures are for ceftriazone and ampicillin would be expected is important to place this study into context).

Reviewer #2: This study reveals very interesting antibiotic combinations against Enterococci in vivo for medical field. The manuscript is well written and the material and methods are sounds. I suggest the authors to use the latest version of CLSI.

6. PLOS authors have the option to publish the peer review history of their article (what does this mean?). If published, this will include your full peer review and any attached files.

Reviewer #1: No

Reviewer #2: No

---

## [Author Response · Author response to Decision Letter 0]

3 Nov 2020

Reviewer #1: interesting pkpd study of combination therapy for Enterococcus. The authors applied proven pkpd concepts and the study is well done. My only comment to the authors is that I agree that it is clear that combination effect is most pronounced in the middle of the exposure curve, and perhaps not surprisingly as they note that at the extremes of exposures, the addition of a second agent is unlikely a priori to make too much of a difference. so with that in mind, it begs a clear question here of where on the ampicillin exposure curve are the mice relative to human exposures, so that one can put this all in clinical context. For example, are the ampicillin exposures, given it is given IV at high doses and frequent administration, already maxing out at T>MIC >50%, and if so then this model does not explain why ceftriaxone may be helpful as when ampicillin exposures are >50% you max out the effect.

We agree with the reviewer that it is necessary to determine where in the ampicillin exposure curve are the mice relative to humans to be able to put the results in clinical context. For this end, we simulated the human pharmacokinetics of two doses of ampicillin, one for soft-tissue infections (500 mg every 6 hours) and one for endocarditis (2000 mg every 4 hours), using the population model published by Soto et al. (Clinical Pharmacol 2014; 77(3):509-21). This model was developed for ampicillin-sulbactam, but it is useful because sulbactam does alter ampicillin pharmacokinetics. New sections of Human AMP PK/PD simulations were added to Methods (lines 197-205 of the revised version), Results (lines 279-285), and Supporting information (Table S2, Figures S4 and S5).

After obtaining the PK profile we estimated the AMP fT>MIC for a range of minimal inhibitory concentrations from 0.5 to 256 mg/L (new S2 Table) to be able to compare murine and human exposures. With the lower dose, we found that for strains with MIC up to 2 mg/L (the majority of isolates: 94% of 13,858 reported in EUCAST international MIC distribution, https://mic.eucast.org) the fT>MIC is at least 61.8%, an exposure leading to maximal efficacy and therefore without benefit from CRO combination. Notwithstanding, for more resistant strains (MIC up to 8 mg/L), this dose is located in middle to left region of the curve, where the addition of CRO would turn an ineffective dose into a bactericidal one, without increasing the amount of ampicillin.

With the higher dose, the simulation showed that the fT>MIC was �63% for strains with MICs up to 16 mg/L, suggesting that this dose is located in the right region of the curve, reaching maximal efficacy and not sensitive to CRO potentiation. However, endocarditis is a difficult to treat infection due to factors like poor penetration of some antibiotics into the vegetations and altered metabolic bacterial state, and even a fT>MIC of 100% is not sufficient to achieve cure (see below Carbon C. 1993). Then, another PK/PD index is required, but unfortunately it has not been determined yet. Notwithstanding, the data from the rabbit endocarditis model suggest that for �-lactams, plasma concentration needs to be all the time above 10 times the minimal bactericidal concentration (MBC). Due to the high MBC of ampicillin among Enterococci (>128 mg/L), even the maximal ampicillin dose (12 g/day) will fail to reach those concentrations, indicating that this dose is actually located in the left region of the curve (yet to be determined experimentally), where CRO potentiation would be feasible (a hypothesis worth testing). Two new paragraphs were included in the Discussion (lines 313-334) with two additional supporting references: 

• Carbon C. Experimental endocarditis: a review of its relevance to human endocarditis. J Antimicrob Chemother. 1993;31

• Pericas JM, Garcia-de-la-Maria C, Brunet M, Armero Y, Garcia-Gonzalez J, Casals G, et al. Early in vitro development of daptomycin non-susceptibility in high-level aminoglycoside-resistant Enterococcus faecalis predicts the efficacy of the combination of high-dose daptomycin plus ampicillin in an in vivo model of experimental endocarditis. J Antimicrob Chemother. 2017;72.

The only other comment I have is that another factor that is not taken into account in these studies is time. These are 24 h studies, you can only have so much antibacterial effect in 24h of time, and thus you might see potentiating or synergistic activities outside this 24h window. Many studies have shown that when you compare short and longer duration studies, you have a very similar ED50 but significantly change the Emax and slope as you get much deeper kill with longer duration. This is a good example that it does significantly depend where on the exposure curve you are located when translating over the animal model hill curves to clinical medicine (and again why elucidating where on the curve the human exposures are for ceftriaxone and ampicillin would be expected is important to place this study into context).

We agree that a longer treatment may lead to higher bactericidal effect. With the highest doses tested in combination, the mean bacterial count after 24 hours was around 3 log10 CFU/g, indicating that there is room for an additional 1 log10 kill (the model’s limit of detection is 2 log10 CFU/g). However, even with this increased killing, the combination would not reach the 2 log10 CFU/g threshold necessary for synergism and still be considered potentiation. Further studies in other models would be required to expand the understanding of the in vivo exposure-response relationship, specially the impact of treatment duration.

Additionally, we chose the standard murine thigh infection model with 24 hours treatment (see Zak and Sande (Ed.). Handbook of animal models of infection, chapter 15, Academic Press 1999), because it has been thoroughly validated and the PK/PD indices derived from it have been shown to accurately predict findings in humans (see Ambrose PG et al. Pharmacokinetics-Pharmacodynamics of antimicrobial therapy: it’s just not for mice anymore. Clinical Infectious Diseases 2007; 44(1):79-86).

Reviewer #2: This study reveals very interesting antibiotic combinations against Enterococci in vivo for medical field. The manuscript is well written and the material and methods are sounds. I suggest the authors to use the latest version of CLSI.

Thank you for the observation. The latest version of CLSI (2020) was used and the reference was updated: Performance standards for antimicrobial susceptibility testing. 30th edition. CLSI supplement M100.: Wayne, PA: Clinical and Laboratory Standards Institute; 2020.

---

## [Decision Letter · Decision Letter 1]

20 Nov 2020

A new pharmacodynamic approach to study antibiotic combinations against Enterococci in vivo: application to Ampicillin plus Ceftriaxone

PONE-D-20-19379R1

Dear Dr. Rodriguez,

We’re pleased to inform you that your manuscript has been judged scientifically suitable for publication and will be formally accepted for publication once it meets all outstanding technical requirements.

Kind regards,

Yung-Fu Chang

Academic Editor

PLOS ONE

Additional Editor Comments (optional):

Reviewers' comments:

Reviewer's Responses to Questions

**Comments to the Author**

1. If the authors have adequately addressed your comments raised in a previous round of review and you feel that this manuscript is now acceptable for publication, you may indicate that here to bypass the “Comments to the Author” section, enter your conflict of interest statement in the “Confidential to Editor” section, and submit your "Accept" recommendation.

Reviewer #2: All comments have been addressed

2. Is the manuscript technically sound, and do the data support the conclusions?

Reviewer #2: Yes

3. Has the statistical analysis been performed appropriately and rigorously? 

Reviewer #2: Yes

4. Have the authors made all data underlying the findings in their manuscript fully available?

Reviewer #2: Yes

5. Is the manuscript presented in an intelligible fashion and written in standard English?

Reviewer #2: Yes

6. Review Comments to the Author

Reviewer #2: (No Response)

7. PLOS authors have the option to publish the peer review history of their article (what does this mean?). If published, this will include your full peer review and any attached files.

Reviewer #2: **Yes: **Jiabin Li

---

## [Editor Report · Acceptance letter]

25 Nov 2020

PONE-D-20-19379R1 

A new pharmacodynamic approach to study antibiotic combinations against Enterococci *in vivo*: application to Ampicillin plus Ceftriaxone 

Dear Dr. Rodriguez:

I'm pleased to inform you that your manuscript has been deemed suitable for publication in PLOS ONE. Congratulations! Your manuscript is now with our production department. 

Kind regards, 

on behalf of

Dr. Yung-Fu Chang 

Academic Editor

PLOS ONE